# Cardiolipin Antibody: A Potential Biomarker for Depression

**DOI:** 10.3390/jpm12111759

**Published:** 2022-10-24

**Authors:** Renzo Costa, Evangelia Fatourou, Debra Hoppensteadt, Jawed Fareed, Angelos Halaris

**Affiliations:** 1Department of Psychiatry and Behavioral Neurosciences, Loyola University Chicago Stritch School of Medicine, Maywood, IL 60153, USA; 2Department of Psychiatry, Icahn School of Medicine at Mount Sinai/Elmhurst Hospital Center, New York, NY 10029, USA; 3Department of Pathology, Laboratory Medicine, Loyola University Medical Center, Maywood, IL 60153, USA

**Keywords:** Cardiolipin antibody, major depressive disorder, escitalopram, quetiapine

## Abstract

**Background**: Inflammation plays a pivotal role in the etiopathology of Major Depressive Disorder (MDD), at least in a subset of patients. It is crucial to first establish which specific inflammatory biomarkers are of clinical utility. Anti-cardiolipin antibody (aCL IgM) is an inflammatory marker that has the potential to be such a candidate but there are insufficient studies to confirm this potential. **Objective**: To investigate the baseline titer level and the longitudinal progression of plasma titers of aCL IgM in MDD subjects receiving antidepressant therapy in comparison to healthy control (HC) subjects; to determine if changes in aCL IgM plasma titers correlate to changes in depressive symptoms; and, to ascertain if baseline aCL IgM plasma titers could predict treatment response. **Methods**: Forty-eight medically healthy outpatients diagnosed with MDD were enrolled in one of two groups in two sequentially conducted clinical trials. In Group-E, patients received a 12-week regimen of escitalopram (*n* = 20). Those in Group-Q received a 12-week regimen of quetiapine (*n* = 28). The main outcome measure was plasma aCL IgM titers, the Hamilton Rating Scale for Depression (HAM-D17) and the Hamilton Rating Scale for Anxiety (HAM-A). There were 16 HC subjects. **Results**: When Group-Q and Group-E participants were grouped together (*n* = 48), MDD subjects had an elevated baseline aCL IgM (19.9 μg/mL) compared to HC subjects (8.32 μg/mL) (*p* = 0.006). aCL IgM correlated significantly with HAM-D17 scores at baseline in MDD subjects (*p* = 0.0185, r = 0.296). Examining the individual groups, Group-Q MDD patients had a significantly elevated baseline aCL IgM (*p* = 0.008) while Group-E’s MDD patients did not. On the other hand, only Group-E MDD patients showed a significant correlation at baseline between aCL IgM and HAM-A score (*p* = 0.0392, r = 0.4327); they also showed a significant inverse correlation between week 12 HAMD-17 Item #10 (Anxiety, Psychic) and week 12 aCL IgM titer (*p* = 0.0268, r = −0.5516). **Conclusions**: MDD patients had significantly higher plasma titers of aCL IgM when compared to HC subjects. Moreover, at baseline, the higher the aCL IgM titer, the higher the depression severity, as measured by HAMD-17 score. However, this study did not demonstrate that aCL IgM titers changed significantly throughout a 12-week course of antidepressant treatment and revealed no correlation between changes in depressive symptoms and changes in aCL IgM titers. Baseline aCL IgM could not predict treatment response. We conclude that, despite lacking predictive ability as regards treatment response, plasma titers of aCL IgM have a diagnostic potential in MDD that necessitates further exploration.

## 1. Introduction

The FDA has approved over 25 medications to treat Major Depressive Disorder (MDD) [1], with the most recent antidepressant medication, esketamine, being approved by the FDA for treatment resistant MDD (TRMDD) in 2019 [2]. Despite recent advances in unraveling the etiopathology of depressive illness and the introduction of novel anti-depressants, MDD afflicts a large percentage of the population worldwide along with high morbidity, mortality, and socioeconomic costs. Remission rates are estimated to be around 35% [3] and about one-third of MDD patients fail to respond to conventional antidepressants [4]. This is not surprising, as we have yet to identify the biological underpinnings of this psychiatric disorder. A promising avenue of exploration is the role of inflammation in the pathophysiology of depression, as revealed by a panel of peripheral biomarkers indicative of chronic immune system activation [5]. MDD patients have been shown to have increased pro-inflammatory mediators and acute-phase reactants [4]. Administration of inflammatory cytokines induces depressive symptoms in non-depressed participants and inhibition of inflammatory signaling pathways reduces symptoms of depression in both MDD patients and patients with chronic, inflammatory diseases [4]. Furthermore, targeting inflammation to produce an anti-inflammatory effect in medicated, depressed patients, has been shown to augment the anti-depressant effect [6]. In treatment resistant bipolar depressed patients, Edberg et al. found that escitalopram co-administered with celecoxib amplified antidepressant response in these patients more so than escitalopram and placebo combination [7].

It is crucial to first establish which specific inflammatory biomarkers play a role in MDD. The precise panel of biomarkers has yet to be validated. C-reactive protein (CRP) is one of the most consistently identified inflammatory biomarkers in many depressed patients [8]. Peripheral blood levels of IL-1β, IL-6, and TNF are also strong candidates, based on a meta-analysis of the literature, but there are many biomarkers left to explore [4]. Anti-cardiolipin (aCL) antibody is an inflammatory marker that has the potential to be a valuable candidate. However, there are insufficient studies to confirm this potential. Cardiolipin is a phospholipid, belonging to a class of molecules important for neurotransmitter signal transduction and neurodevelopment. Dysregulation of phospholipids contributes to CNS dysfunction [9]. Antiphospholipid Syndrome (APS) is one of the clearest clinical examples of the neuropsychiatric effects of phospholipid dysregulation. APS is an autoimmune disease characterized by a hypercoagulable state that results in recurrent arterial and venous thrombosis and/or recurrent miscarriages. The presence of antiphospholipid autoantibodies (directed towards, for example, beta 2-glycoprotein I (B2 GPI), prothrombin, cardiolipin, phosphatidylserine, tissue plasminogen activator, plasmin, annexin A2 and thrombin) is required for establishing the diagnosis. There is a growing, albeit limited, body of evidence indicating psychiatric symptoms (including mania, psychosis, anxiety, and depression) as one of the earliest manifestations of this disease [10]. APS patients have higher depressive symptoms and, inversely, depressed patients have higher antiphospholipid antibody titers than healthy control (HC) subjects [9]. 

We make the assumption that a patient need not have diagnosable APS for antiphospholipid dysregulation to have neuropsychiatric ramifications. Rather, APS can be thought of as one end of a spectrum—a spectrum that points towards increased antiphospholipid antibodies as a potential factor in depression. There is evidence in support of this assumption. For example, Maes et al., (1993) observed that depressed patients had higher antiphospholipid antibody titers in comparison to HC subjects, but not nearly as high as in patients with systemic lupus erythematosus or in patients displaying overt complications of APS (thrombocytopenia, fetal loss, etc.) [11]. aCL antibodies, in particular, have consistently been associated with neuropsychiatric complications, in comparison to other antiphospholipid antibodies. Schwartz et al., (1998) found that 8 out of 34 medication-naïve patients, recently admitted for acute psychosis, had increased titers of aCL IgG, despite no prior diagnosis of APS or any other autoimmune disorder [12]. Nineteen years later, a meta-analysis found that patients with elevated aCL antibodies were 3.45 times more likely to present with dementia than healthy controls [13]. A 2019 systematic review then reported a positive association between aCL antibodies and cognitive impairment [14].

As such, this study focuses on aCL IgM. The primary goals were to investigate the progression of aCL IgM plasma titers in MDD patients throughout a course of antidepressant pharmacotherapy in comparison to HC subjects, and to determine if changes in aCL IgM correlated to changes in depressive symptoms. Our secondary goal was to ascertain if baseline aCL IgM could predict treatment response.

## 2. Methods

### 2.1. Study Population

Institutional Review Board (IRB) approval for this study was obtained through the IRB of Loyola University Medical Center. It was conducted according to the principles of the Declaration of Helsinki. Participation in the study required that candidates be capable of understanding the nature of the study and giving written informed consent. Male and female patients, between the ages of 20 to 65 years, had to be diagnosed with MDD, as defined by the Diagnostic and Statistical Manual of Mental Disorders-Fourth Edition (DSM-IV), and be physically healthy. In order to be included in the study, the depressive episode had to be currently ongoing, and of at least one month’s duration; the patient had to be free of psychopharmacological treatment for four weeks immediately prior to participation; and, a minimum score of 18 on the 17-item Hamilton Depression Scale (HAM-D17) was required. Candidates who were already receiving psychopharmacological treatment underwent a 4-week washout period with close monitoring for possible significant worsening of depression or suicidality. Candidates who had previously failed to respond to at least two antidepressant drugs, administered in commonly prescribed doses and for an adequate length of time, were considered to have “treatment resistant MDD” (TRD), but they were not excluded. Exclusion criteria included co-morbidity with other DSM-IV Axis I disorders, except for MDD co-morbid with Generalized Anxiety Disorder (GAD); active suicidality; history of smoking or substance abuse in the preceding 6 months; comorbidity with inadequately treated chronic medical illnesses (e.g., diabetes, hypertension); active inflammation including gum disease; and actively undergoing any other therapeutic interventions (e.g., psychotherapy). They were also excluded if they were taking Cytochrome P450 (e.g., P4503A4) inducers (e.g., ketoconazole, itraconazole, fluconazole, erythromycin, clarithromycin, troleandomycin, indinavir, nelfinavir, ritonavir, fluvoxamine, saquinavir, phenytoin, carbamazepine, barbiturates, rifampin, St. John’s Wort, and glucocorticoids). Female subjects were excluded if they were pregnant, lactating, or on oral contraceptives; if they were sexually active, they had to be practicing reliable contraception for the duration of the study.

### 2.2. Study Design

Screening included post-fasting blood samples that ensured a healthy status (complete blood count, complete metabolic panel, lipid profile, thyroid function, urinalysis, and pregnancy test), completion of the Mini International Neuropsychiatric Interview (MINI), and a Family History Questionnaire.

Upon successful completion of screening, all subjects had their blood drawn for inflammatory biomarkers and completed the Hamilton Rating Scale for Depression (HAM-D17) and Hamilton Rating Scale for Anxiety (HAM-A) during the baseline visit.

MDD patients were enrolled in one of two treatment protocols in two consecutively run studies. Group-E received a 12-week regimen of escitalopram 20–40 mg/day (*n* = 20) [5]. Group-Q received a 12-week regimen of quetiapine, up to 150 mg/day, as tolerated (*n* = 28) [15]. No other form of antidepressant therapy was given during these 12 weeks. Follow-up visits were scheduled for weeks 1, 2, 4, 8, and 12. During these visits, blood draws were repeated at weeks 4 and 8, and the patients completed the same scales that were administered at the baseline visit. At each visit, tolerability assessments were made based on the patient’s subjective symptoms and their willingness to remain in the study. 

For a more detailed description of the study design and outcome for study Group-E and study Group-Q, please refer to Halaris et al. (2015) [5] and Arisoy et al. (2019) [15], respectively. Furthermore, the rationale for studying quetiapine monotherapy for MDD is described in detail in the latter publication. 

### 2.3. Study Participants

Using the inclusion and exclusion screening criteria, 77 subjects were deemed eligible for these studies and successfully completed baseline assessments. Four subjects dropped prior to receiving the first dose of medication and the remaining 73 began treatment. Forty-eight subjects completed the week-8 assessments. Forty-four completed the week-12 assessments.

To be considered a completer, a subject had to complete at least the first 8 weeks of treatment. For completers who withdrew from the study at week 8 or later, their last observations were carried forward to week 12 (LOCF). Subjects that experienced intolerable side effects, or did not respond to the medications, were withdrawn from the study and treated with another antidepressant at the investigator’s discretion.

Patients who experienced a 50% reduction in their HAM-D17 score from baseline to the end of the study were considered treatment “responders”. Those whose end-of-study HAM-D17 scores decreased to 7 or less were considered to be “remitters”. Those whose HAM-D17 scores underwent a 50% reduction, but did not decrease to 7 or less, were considered “partial responders”. Those whose end-of-treatment score did not reach a minimum of a 50% reduction were considered to be “non-responders”.

### 2.4. Healthy Control Subjects

Healthy control (HC) subjects were recruited via word of mouth and flyers on the Loyola University Medical Center campus. Enrollment for HC subjects occurred alongside MDD patient enrollment. Eligibility criteria for HC subjects included a HAM-D17 score < 5, a Beck Depression Inventory score < 5, and the absence of medical illness, mental illness, active inflammation including gum disease, substance use, and mental illness or substance use amongst first degree relatives. Female candidates who were pregnant, lactating, or taking oral contraceptives were excluded. HC subjects underwent the same screening assessments by the same clinician investigators as the MDD patients. If the HC subjects’ screening results were within normal range and written informed consent was obtained, they were deemed eligible to participate in the study and a baseline visit was scheduled. The baseline visit for HC subjects was identical to the baseline visit for MDD patients. After the baseline visit was completed, no further blood draws were done on HC subjects, given the reasonable supposition that the targeted biomarkers remain fairly stable in the absence of any illness or stressful life events. After systematic attempts to match the ages of HC subjects to MDD patients, 16 HC subjects were included in the study.

### 2.5. Biochemical Analyses

For both HC subjects and MDD patients, antecubital venous blood was drawn between 09:00 and 10:00 hours following an overnight fast before their scheduled visit. All participants were to abstain from aspirin, antihistamines, acetaminophens, vitamins C or E, sleeping pills, caffeinated beverages, physical exertion, or tobacco products for several hours prior to blood draw. Once collected, blood samples were separated into plasma and serum samples and were immediately stored at −80 °C. The “Evidence Investigator™” Biochip array system from Randox Technologies was used to measure plasma concentrations of cytokines, chemokines, and growth factors. This system had been previously validated in preliminary studies, in which results were found to be analogous to results obtained by ELISA for each of the individual parameters. ELISA was used to measure plasma titers of aCL IgM.

### 2.6. Statistical Analysis

GraphPad Prism version 8.0.2 was used for the statistical analyses. Pearson correlation was used to analyze the relationship between aCL IgM titers and rating scales, while *t*-test was used to compare aCL IgM titers at different time points and between different groups.

## 3. Results

### 3.1. Combined Groups

When Group-Q and Group-E MDD patients were grouped together (*n* = 48), there was a statistically significantly elevated baseline aCL IgM plasma titer (19.9 μg/mL) compared to HC subjects (8.32 μg/mL) (*p* = 0.006) (Figure 1). The aCL IgM correlated significantly with HAM-D17 scores at baseline in MDD subjects (*p* = 0.019, r = 0.296) (Figure 2). There was no statistically significant relationship between aCL IgM and HAM-A scores at baseline. More specifically, HAM-D17 item #7 (Work and Activities), item #10 (Anxiety, Psychic), and item #11 (Anxiety, Somatic) each failed to exhibit a significant correlation with aCL IgM at baseline.

Titers of aCL IgM did not change significantly between baseline and week 8, baseline and week 12, or week 8 and week 12. Week 12 aCL IgM did not correlate with either week 12 HAM-D17 scores or week 12 HAM-A scores. Additionally, HAM-D17 items #7, #10, and #11 each did not correlate with end-of-treatment aCL IgM.

With regard to treatment response, there was no statistically significant change in aCL IgM between baseline and week 8 or baseline and week 12 in any of the three response groups (responders, non-responders, or remitters). 

### 3.2. Group-Q

Titers of aCL IgM at baseline were statistically significantly higher in MDD patients in comparison to HC subjects’ (*p* = 0.0005); mean aCL IgM was 23.97 μg/mL for MDD patients and 8.322 μg/mL for HC subjects (Figure 3). At baseline, there was no correlation between HAM-D17 scores and aCL IgM, nor was there any correlation between HAM-A scores and aCL IgM. More specifically, HAM-D17 items #7, #10, and # 11 each failed to exhibit a significant correlation with aCL IgM at baseline. 

When performing a time course analysis, we found that aCL IgM did not change significantly between baseline and week 8, baseline and week 12, or week 8 and week 12. Week 12 aCL IgM did not correlate with either week 12 HAM-D17 scores or week 12 HAM-A scores. Additionally, HAM-D17 items #7, #10, and #11 each did not correlate with week 12 aCL IgM.

When MDD patients were separated into treatment response groups, there was no statistically significant change in aCL IgM between baseline and week 8 or baseline and week 12.

### 3.3. Group-E

MDD patients did not differ significantly in any baseline variables when compared to HC subjects (Figure 4). However, within the MDD group at baseline, there was a statistically significant correlation between aCL IgM and HAM-A scores (*p* = 0.039, r = 0.433) (Figure 5). HAM-D17 scores did not correlate with aCL IgM at baseline. More specifically, baseline aCL IgM did not exhibit a significant correlation with either HAM-D17 items #7, #10, or # 11.

When performing a time course analysis, we found that aCL IgM did not change significantly between baseline and week 8, baseline and week 12, or week 8 and week 12. Week 12 aCL IgM did not correlate with either week 12 HAM-D17 scores or week 12 HAM-A scores. Week 12 aCL IgM did not correlate with either HAM-D17 item #7 or #11. However, week 12 aCL titer did have a significant inverse correlation with week 12 HAM-D17 item #10 (*p* = 0.027, r = −0.552) (Figure 6).

When MDD patients were separated into treatment response groups, there was no statistically significant change in aCL IgM between baseline and week 8 or baseline and week 12.

### 3.4. Intergroup (Group-Q vs. Group-E) Differences

The two MDD groups differed significantly by sex, with 55.32% of Group-Q being female versus 78.05% of Group-E (*p* = 0.0416), and age (Group-Q mean age = 43.79, Group-E mean age = 37.56, *p* = 0.0169). At baseline, MDD patients in Group-Q had a significantly elevated aCL IgM when compared to Group-E (*p* = 0.008).

When comparing the change in Group-E MDD patients’ aCL IgM from baseline to week 12 versus the change in Group-Q MDD patients’ aCL IgM from baseline to week 12, there was no statistically significant difference.

## 4. Discussion

We sought to identify any discernible patterns of aCL IgM plasma titers in MDD patients before and after short-term antidepressant drug treatments. MDD patients had significantly higher plasma titers of aCL IgM when compared to HC subjects at baseline. Moreover, at baseline, elevations in aCL IgM titers correlated significantly with increased depression severity scores. We did not detect any significant change in aCL IgM titers throughout the 12-week course of antidepressant treatment with either of the wo agents we used, nor did we detect any correlation between changes in depressive symptoms and changes in aCL IgM titers. We postulate that 12 weeks of treatment may not be sufficient to observe any measurable differences in inflammatory biomarkers in the treatment of MDD. The anti-inflammatory effects of antidepressants may become measurable only after depressive symptoms have been stabilized over a longer time frame of maintenance treatment. A potential timeline is displayed below.

Increased inflammatory markers ←→ increased HAM-D17→ treatment → decreased HAM-D17 → decreased inflammatory markers

We recommend that future studies on inflammation and depression, including time series analyses of plasma aCL IgM titers, extend the timeframe for observation beyond the conventional clinical trial timeline (8–12 weeks) in order to establish sequential effects of treatment on the proinflammatory state of the patient as revealed in levels of inflammatory markers. Failure to do so may lead to misinterpretation of the obtained results. With respect to aCL IgM, it is also critical to establish whether elevated titers persist in spite of sustained remission of the psychiatric disorder being treated. Delineating the chronological relationship between aCL antibody and psychiatric disorders may provide insight into the potentially causative role for this inflammatory marker. Since studies have demonstrated aCL antibody’s role in microthrombi formation within the CNS [16] along with its ability to disrupt the blood-brain-barrier [17,18] and directly damage neural cells [19,20,21], further investigation into aCL antibody’s relationship with neuropsychiatric symptoms is warranted. In view of evidence that aCL IgM has been associated with dementia [13,14], psychosis [22], and cognitive decline [14], such studies are urgent.

Considering the two MDD groups in the present study, interesting results have emerged. At baseline, Group-Q’s MDD patients had significantly higher aCL IgM titers than HC subjects. This was not evident in Group-E. On the other hand, only Group-E’s MDD patients showed a significant correlation at baseline between aCL IgM titer and anxiety severity, as measured by HAM-A. We speculate that the demographic differences in the groups might account for the differences in results. There was a statistically higher percentage of female patients in Group-E (78.05%) compared to Group-Q (55.32%). Based on the thorough review by Bekhbat and Neigh, there are sex-specific differences in neuroinflammation and consequent neuropsychiatric symptoms [23]. Different inflammatory biomarkers can affect males and females differently. For example, Liukkonen et al. reported that C-reactive protein to be associated with depression in men, but not in women [24]. One should also consider the sexually dimorphic presentations of depression. As Marcus et al. reported in their evaluation of the STAR*D (Sequenced Treatment Alternatives to Relieve Depression) multicenter trial, women are more likely than men to manifest depression with symptoms of anxiety [25]. Taken together, this line of reasoning suggests aCL antibodies have a stronger association with MDD in men, and anxiety symptoms in women. Furthermore, Group-Q patients were, on average, older than Group-E patients. With age comes dysregulation of the immune response, both at a molecular and systemic level; this, in turn, results in a chronic systemic proinflammatory state [26]. An older cohort could also contribute to the higher baseline aCL IgM titer found in Group-Q.

Miller and Raison suggested that future research on the role of inflammation in depression should pay particular attention to measures of anhedonia and anxiety, as “inflammation targets neurocircuits in the brain that regulate motivation and reward as well as anxiety, arousal and alarm” [4]. Following this suggestion, specific HAM-D17 items were analyzed: item #7 (Work and Activities) as a proxy for anhedonia, and both items #10 (Anxiety, Psychic) and #11 (Anxiety, Somatic) as proxies for anxiety. We found one significant correlation with these items. Week 12 item #10 had a significant inverse correlation with week 12 aCL IgM titer in Group-E. Overall, this anxious group did not have elevated plasma titers of aCL IgM at baseline. These results run contrary to Miller and Raison’s suggestion. It has yet to be determined if aCL IgM plays a different role in non-anxious-type depression, as opposed to anxious-type depression, and, possibly, treatment-resistant depression.

## 5. Conclusions

We found that MDD patients tend to have higher plasma titers of aCL IgM when compared to HC subjects. In these MDD patients, titers of aCL IgM did not change significantly throughout a 12-week course of antidepressant treatment, regardless of treatment response or pharmacologic agent administered. We conclude that plasma titers of aCL IgM have diagnostic potential, despite lacking ability to predict treatment response. Researchers should consider inclusion of aCL IgM in future inflammatory biomarker panels for MDD. Future studies should also consider extending their observational period beyond the 12-week trial used in this study, in order to determine whether titers normalize after a longer time interval. Lastly, we recommend future studies acknowledge sex and age as potential confounding variables, and account for them, as each of these variables impacts inflammatory processes in the context of MDD.

## Figures and Tables

**Figure 1 jpm-12-01759-f001:**
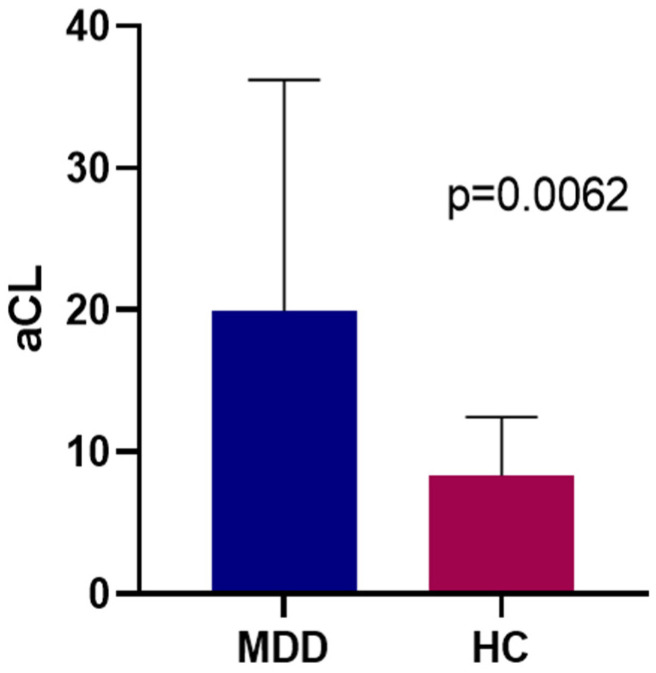
aCL IgM titers at baseline for all MDD patients (Mean = 19.93 μg/mL) and HC (Mean = 8.322 μg/mL).

**Figure 2 jpm-12-01759-f002:**
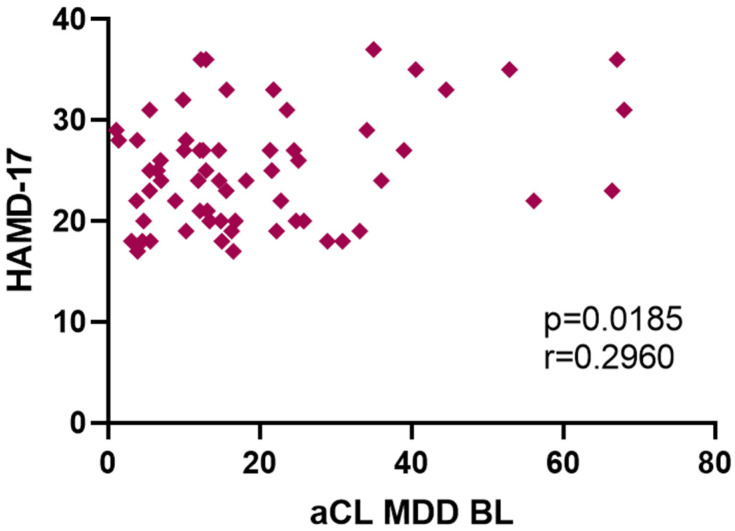
aCL IgM titers of all MDD patients in correlation with HAMD-17 at baseline.

**Figure 3 jpm-12-01759-f003:**
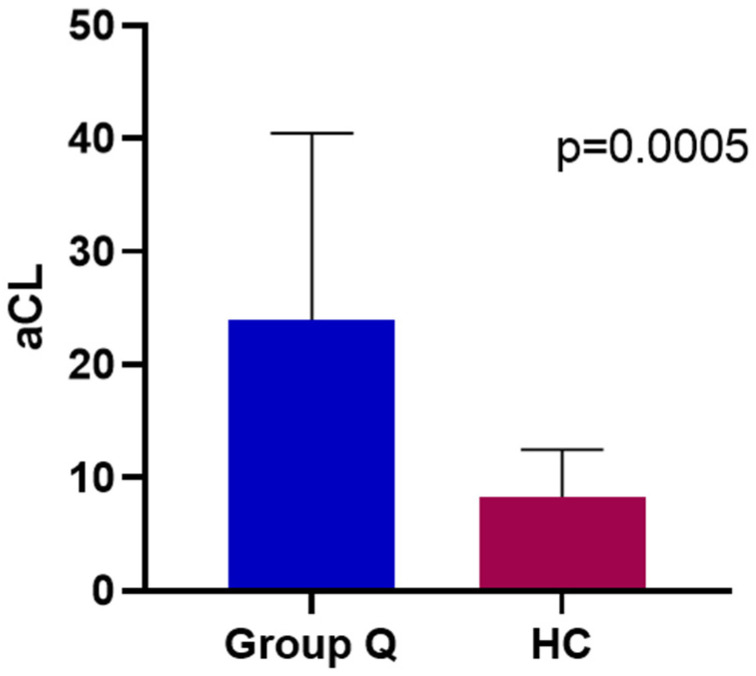
aCL IgM titers at baseline for Group Q MDD patients (Mean = 23.97 μg/mL) and HC (Mean = 8.322 μg/mL).

**Figure 4 jpm-12-01759-f004:**
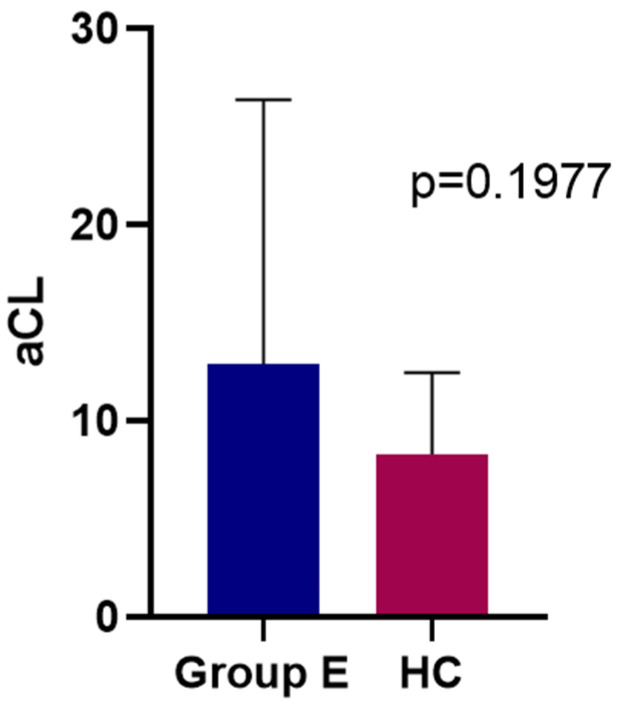
aCL IgM titers at baseline for Group E MDD patients (Mean = 12.9 μg/mL) and HC (Mean = 8.322 μg/mL).

**Figure 5 jpm-12-01759-f005:**
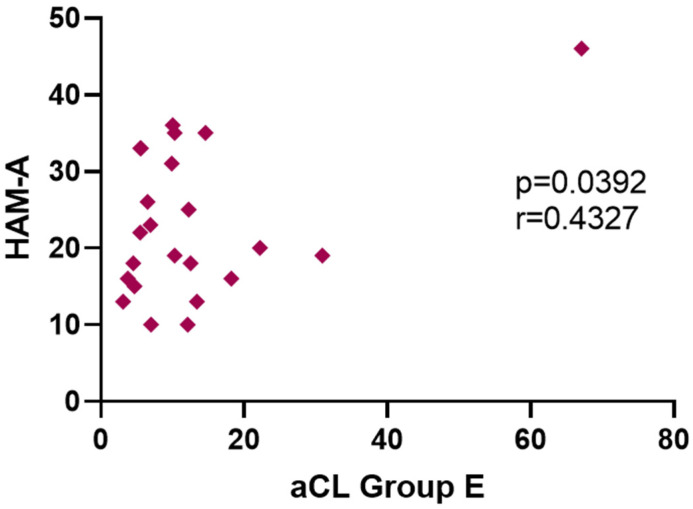
aCL IgM titers of Group E MDD patients in correlation with HAM-A baseline.

**Figure 6 jpm-12-01759-f006:**
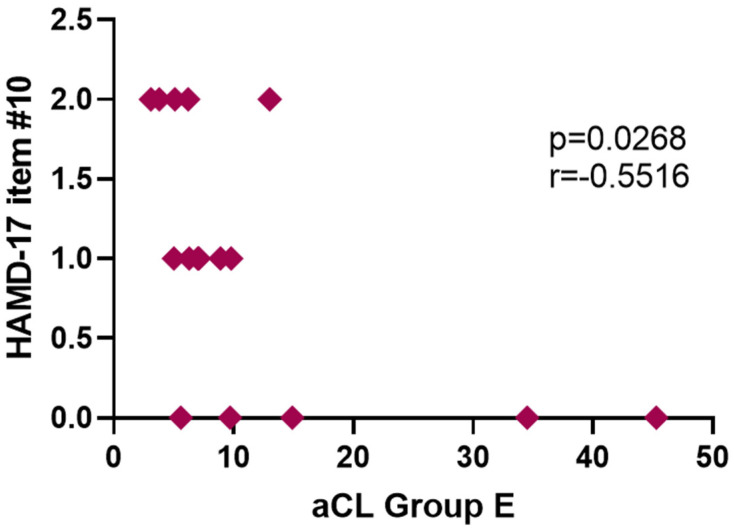
aCL IgM titers for Group E MDD patients in correlation with item #10 of HAMD-17 (Anxiety, Psychic) at week 12.

## Data Availability

The data presented in this study are available on request from the corresponding author. The data are not publicly available due to confidentially reasons.

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
