# Peer review of "Cardiolipin Antibody: A Potential Biomarker for Depression"

_jpm, 2022, doi:10.3390/jpm12111759_

Round 1

Reviewer 1 Report

The manuscript discusses the titers of antibodies to cardiolipin in depressed patients before and after antidepressant therapy   in comparison with a group of clinically healthy individuals. The authors suggested that it is not necessary for a patient to have diagnosable APS for antiphospholipid dysregulation to have neuropsychiatric consequences.

Recently, antiphospholipid syndrome has attracted attention in many fundamental and clinical aspects, including somatic and neuropsychiatric pathologies. The search for adequate markers of depression remains an actual problem. The authors found that antibodies to cardiolipin can be a marker of depression and the depth of pathology, but not a marker of pharmacological antidepressant activity.

The list of references to scientific literature contains approximately 30% of sources published in the last 5 years.

Author Response

We thank the reviewer for the positive evaluation of our manuscript.

Reviewer 2 Report

Costa et al. investigated the potential of aCL as a biomarker for predicting MDD severity and treatment effect. The authors found that aCL IgM plasma titer increased significantly in MDD subjects compared to controls and correlated significantly with baseline MDD severity, especially in subjects on escitalopram. The authors did not find treatment related changes in aCL IgM titer after 12 weeks of treatment. The paper is well-written and clearly illustrated.

Minor issues:

1. In Method-Biochemical Analyses line 1, was the blood drawn 9-10 hours after overnight fasting or drawn between 9:00 and 10:00 a.m? Please clarify.

2. For correlation plots (Figure 2, 5, and 6), please also add correlation coefficient r to the plots.

Author Response

1. Blood was drawn between 9 and 10 am after overnight fasting.

2. The correlation r's were added to the figure legends as the entire graphs would have had to be redrawn which would taken a lot longer that then 3-day turnaround time required for the resubmission of the the revised manuscript.